# Preparative Fractionation of Brazilian Red Propolis Extract Using Step-Gradient Counter-Current Chromatography

**DOI:** 10.3390/molecules29122757

**Published:** 2024-06-09

**Authors:** Begoña Gimenez-Cassina Lopez, Maria Cristina Marcucci, Silvana Aparecida Rocco, Maurício Luís Sforça, Marcos Nogueira Eberlin, Peter Hewitson, Svetlana Ignatova, Alexandra Christine Helena Frankland Sawaya

**Affiliations:** 1Faculty of Pharmaceutical Science, State University of Campinas (UNICAMP), Campinas 13083-871, SP, Brazil; begogcl@gmail.com; 2Department of Biosciences and Oral Diagnosis, Institute of Science and Technology, São Paulo State University (ICT-UNESP), São José dos Campos 12245-000, SP, Brazil; cristina.marcucci@unesp.br; 3Brazilian Biosciences National Laboratory (LNBio), Brazilian Center for Research in Energy and Materials (CNPEM), Campinas 13083-970, SP, Brazil; silvana.rocco@lnbio.cnpem.br (S.A.R.); mauricio.sforca@lnbio.cnpem.br (M.L.S.); 4MACKGRAPHE—School of Engineering, Mackenzie Presbyterian University, São Paulo 01302-907, SP, Brazil; mneberlin@gmail.com; 5Department of Chemical Engineering, College of Engineering, Design and Physical Sciences (CEDPS), Brunel University London, Uxbridge UB8 3PH, UK; peter.hewitson@brunel.ac.uk

**Keywords:** injection strategy, preparative fractionation, retusapurpurin A, 3(*R*)-7-O-methylvestitol, prenylated benzophenone isomers

## Abstract

Propolis is a resinous bee product with a very complex composition, which is dependent upon the plant sources that bees visit. Due to the promising antimicrobial activities of red Brazilian propolis, it is paramount to identify the compounds responsible for it, which, in most of the cases, are not commercially available. The aim of this study was to develop a quick and clean preparative-scale methodology for preparing fractions of red propolis directly from a complex crude ethanol extract by combining the extractive capacity of counter-current chromatography (CCC) with preparative HPLC. The CCC method development included step gradient elution for the removal of waxes (which can bind to and block HPLC columns), sample injection in a single solvent to improve stationary phase stability, and a change in the mobile phase flow pattern, resulting in the loading of 2.5 g of the Brazilian red propolis crude extract on a 912.5 mL Midi CCC column. Three compounds were subsequently isolated from the concentrated fractions by preparative HPLC and identified by NMR and high-resolution MS: red pigment, retusapurpurin A; the isoflavan 3(*R*)-7-O-methylvestitol; and the prenylated benzophenone isomers xanthochymol/isoxanthochymol. These compounds are markers of red propolis that contribute to its therapeutic properties, and the amount isolated allows for further biological activities testing and for their use as chromatographic standards.

## 1. Introduction

Propolis is a resin that bees collect from the barks, buds, and flowers of different plant sources. In the hive, bees use propolis to cover the walls and protect it against insects or climate-related situations such as wind or rain [1,2]. The use of propolis by bees as an antimicrobial has been studied and is one of the reasons why this resin is popularly used in natural product preparations. Red propolis inhibits the growth of Gram-positive bacteria such as *Streptococcus mutans* and *Staphylococcus aureus* [3,4], as well as Gram-negative bacteria such as *Klebsiella pneumoniae*—which is a resistant bacteria—and *Pseudomonas aeruginosa* [5,6]. Its antifungal activity has also been proved, mainly against Candida species [7]. More recently, its antioxidant and cytotoxic activities have also been investigated [8,9].

The composition of red propolis is very complex and depends on the plant sources that bees visit. It contains a large percentage of beeswax, but it also contains sterols, terpenes, and phenolic compounds such as flavonoids. These substances comprise a wide variety of polarities, making the use of preparative solid support-based chromatographic techniques to obtain fractions and isolate compounds from the ethanolic extract of red propolis a tricky task, due to low loadings, column blocking, and irreversible adsorption of target molecules.

Traditionally, red propolis is extracted with 80% ethanol to remove most of the waxes, dead bees, and other residues present in the raw material [8,10]. Further strategies vary between studies. Some used silica columns to obtain fractions, followed by thin-layer chromatography (TLC) [11]. Others employed a Sephadex LH20 column, with further purification using an HPLC C18 column [12], or combinations of these strategies [13]. However, these techniques are time-consuming and often yield only a few milligrams of isolated substances.

When a large amount of the sample requires processing, preparative HPLC is often used as a robust technology, exhibiting high resolving power and high efficiency in the purification of natural products [14,15]. However, a complex natural matrix such as red propolis can block the column when the concentration of the sample is too high, leading to the partial loss of the sample and potentially important compounds, as well as cross-contamination of the fractions [16]. If the target compound is a minor constituent of a complex matrix, the sample concentration and its volume, loaded directly onto a preparative column, often affect the results [17].

Therefore, counter-current chromatography (CCC) emerges as a possible solution to obtain semi-purified fractions for further purification by preparative HPLC. CCC is a solid support-free separation technique that works as a continuous liquid–liquid extraction process, using a solvent system with at least two immiscible solvents. One of these liquid phases is retained inside a column and is referred to as the stationary phase, while the other is continuously pumped through as the mobile phase [18]. The separation principle is based on the partitioning of compounds between these two phases, according to their solubility and polarity. A typical commercial CCC instrument consists of two or three columns rotating in planetary motion, which generates a force field causing continuous mixing and settling of the liquid phases. The solvent system is chosen according to the polarity of the target compounds [18].

CCC has been successfully applied to isolate potentially biologically active compounds from both natural and synthetic mixtures [18,19,20]. The technology was previously used for the metabolite profiling of South Arabian propolis [21]. A total of 800 mg of an ethyl acetate extract was separated with a non-aqueous hexane–acetonitrile (1:1 *v*/*v*) solvent system within 12 h on an 850 mL column. A similar approach was employed for the separation of 711 mg of floral resins from a hexane extract of *Clusia fluminensis* flowers, producing the prenylated benzophenone, clusianone, using an n-hexane–acetonitrile–methanol (2:1.25:0.5 *v*/*v*/*v*) solvent system, within 2 h on a 125 mL column [22]. CCC was also used to isolate nerolidol from 500 mg of the essential oil of *Baccharis dracunculifolia* leaves, using a hexane–methanol–water (5:4:1 *v*/*v*/*v*) system, within 40 min, on a 118 mL column [23]. These results suggest that CCC can be used to fractionate Brazilian red propolis extracts at a preparative scale.

Biological tests for antimicrobial, antioxidant, cytotoxic, or anti-inflammatory activities, or any other possible therapeutic application, require large amounts of high purity compounds for in vitro and in vivo assays. Choosing an appropriate isolation strategy is a key factor. The approach of combining CCC as the initial purification step and preparative HPLC as a final purification step combines the high capacity of CCC and the high-resolution of preparative HPLC.

The aim of this study was to develop a separation strategy for manufacturing large quantities of semi-purified fractions across a wide range of polarity from Brazilian red propolis to allow further studies of these bioactive compounds.

## 2. Results

### 2.1. CCC Method Development

The primary aim of the development of the CCC method was focused on the search for a non-polar solvent system providing a wide range of partition coefficients (KD) for all crude compounds, as well as good sample solubility, with minimal emulsification. The system would be run in the normal phase (NP) elution mode (with an organic non-polar phase as the mobile phase), with waxes eluting with the solvent front. Several solvent systems (Table 1) obtained from the literature [21,22,23], as well as a standard approach used by the authors, were tested. A short HPLC–UV method modified from the authors’ earlier UPLC–MS method [19] was developed for a rapid screening of CCC fractions and was used to establish that the peak of 14.418 min could be used as a KD = 1 marker for method development (Appendix A).

The partition study included hexane- and methanol-based solvent systems, as the former is the best solvent for waxes, and the latter is the best solvent for the crude extract. Testing non-aqueous solvent systems presented in the literature, hexane–methanol–acetonitrile (8:2:5 and 5:0:5 *v*/*v*/*v*), which were previously used for propolis separation [21,22], led the marker compound and the rest of the crude to fully partition into the methanol phase. The polarity range for water containing solvent systems spans from the most non-polar HEMWat 27 (19:1:19:1 *v*/*v*/*v*/*v*) to intermediate HEMWat 17 (1:1:1:1 *v*/*v*/*v*/*v*), with a corresponding KD value range of 0.1–17.5. In HEMWat 25–27 systems, the chosen marker compound tends to partition into the aqueous lower phase, since it contains between 75 to 82% methanol, with 12.5 to 4% water, respectively, whereas the remainder is ethyl acetate. The amount of hexane in the organic upper phase for these systems is 97–95%. The increase in water in the aqueous phase to 17% in HEMWat 23 (4:1:4:1 *v*/*v*/*v*/*v*) or 20% in HMWat 2 (5:4:1 *v*/*v*/*v*) improves the partitioning of the marker compound into the organic upper phase, yielding KD values of 0.6 and 1.2, when calculated for the normal phase elution mode. Further analysis of the KD range showed that HMWat 2 would deliver a better separation for the nonpolar compounds in the crude extract. This result can be explained by the presence of ethyl acetate, which makes HEMWat 23 more polar. The HMWat 2 (5:4:1 *v*/*v*/*v*) solvent system was previously used to isolate a metabolite from the essential oil of one of the plant sources of Brazilian green propolis [23], but obtaining the essential oil is a purification step which was not performed in this study, and green propolis is chemically different from red propolis.

The analytical-scale CCC, run in NP mode (Appendix A) on a 17.4 mL column, with an injection of 40 mg (100 mg/mL in lower phase), delivered several semi-purified fractions, for which indicative HPLC analyses are given in Appendix A. The elution for two column volumes produced a few enriched fractions, confirming that the HPLC and CCC elution order is different, in this case. The marker compound eluting at about KD = 1.2 exhibited the highest purity (above 90%). The extrusion fraction contained too many compounds. The method clearly required further improvement.

The use of a gradient of a mobile phase composition in CCC is particularly useful when applied to a mixture of compounds with a wide polarity range, such as the red propolis extract [24]. The gradient in CCC can be linear or step, often depending on the equipment available or the complexity of the crude. The organic phase of the HMWat solvent system family contains only hexane [25]. Therefore, HEMWat 17 was considered as a step gradient to improve the separation of intermediate and polar compounds in the crude because its organic phase contains 34% ethyl acetate, 63.4% hexane, and 2.6% methanol, whereas HMWat 2 and HEMWat 17 aqueous (stationary) phases are very similar, as they contain mainly methanol and water, which makes the gradient method work well. KD values for HEMWat 17 (Table 2) confirm that compounds with an HPLC elution time of less than 8 min (Appendix A) can be separated into smaller groups in a short run time, including highly polar compounds retained in the stationary phase.

Therefore, an analytical scale separation with a crude loading of 70 mg was performed as a step gradient in the normal phase mode (upper organic phase used as the mobile phase), starting with HMWat 2 from the injection point, followed by a switch to the upper phase of HEMWat 17 at 25 min and extrusion of the column content at 50 min. The 100 mg/mL sample solution was made up in the methanol-containing stationary phase. This method provided a much better separation of compounds retained in the stationary phase.

The volumetric scale-up of the method to the 912 mL Midi instrument, with a 2.0 g sample loading at 50 mL/min, yielded very similar results. When optimizing the sample loading as a part of the CCC method development, sample concentration and volume were considered. However, the increase in the concentration caused the emulsification of the sample solution due to the complex crude composition and the presence of water in the stationary phase. Any emulsification leads to an additional loss of the stationary phase and as a result, decreased fractionation.

To improve the separation, the sample was dissolved in methanol, and post-injection modifications with a stepwise change in the flow rate were employed. Decreasing the mobile phase flow to 1 mL.min^−1^ just after the injection allows the sample to interact with the solvent system, leading to reduced stripping of the stationary phase and therefore, improved separation of the most nonpolar compounds eluting before achieving one column of volume. This strategy was suggested by the authors for a binary mixture separation with one target compound, using hexane-ethyl acetate–ethanol–water (HEEWat 5:2:5:2 *v*/*v*/*v*/*v*) [26]. The ethanol in HEEWat is fully miscible with every solvent in that system and is present in both phases. Therefore, when the sample solution prepared in ethanol was injected, it could easily travel between phases to improve partitioning. In the HMWat system used in this study, methanol is present only in the aqueous lower stationary phase. Injecting the sample in methanol helped to reduce emulsification and lowered the polarity of the stationary phase without affecting mobile phase composition. Also, no preliminary purification of the propolis crude sample was made before this CCC separation, as multiple target compounds were being separated, which required careful balancing between the loading, stationary phase retention, and gradient of the mobile phase composition. Table 3 provides a summary of the tested operating parameters.

The retention of the stationary phase in the CCC column was also assessed due to its higher loss after the method was scaled-up to the preparative DE-Midi. To minimize the loss, the mobile phase flow was reduced to 1 mL·min^−1^ for 10 min after injection for the sample solution to become diluted within the stationary phase, and then it was raised back up to the original flow rate (Table 3).

There are two stages of stationary phase stripping when the step gradient is applied (Figure 1). The first stage occurs during the first 25 min (15 fractions) right after injection, as it takes time for the sample to travel through the 912 mL column. A plateau region on the graph, starting between 0.7 and 0.8 of the column volume, is where most of separation occurs. The optimum initial retention of this region should be 60% or higher. The second stage of the stationary phase stripping occurs after the step gradient switch (fraction 34 onwards, depending on the flow rate). When run at 50 mL·min^−1^ with 2 g of sample and the sample solution obtained in the stationary phase (HMWat 2 lower phase), the overall loss of SP was 55%, before the step gradient (run Midi-01). Changing the sample solvent to methanol in run Midi-02 led to 10% less stripping, which was still below the target of 60%. Decreasing the mobile phase flow rate to 40 mL·min^−1^ and the sample volume to 10 mL (1% of the column volume) allowed the column to achieve 70 and 61% of the SP value for a 100 and 200 mg/mL sample concentration, respectively (runs Midi-03, 04), although this dropped to 52% when the concentration was raised to 400 mL/min (run Midi-05). Finally, 254 mg/mL was found to be the optimum sample concentration in methanol, achieving 60% of the SP retention after the injection stage, showing good reproducibility, as seen for runs Midi-06, 07, and 08. Interestingly, the SP stripping after the step gradient varies slightly, but the slope was identical for each run when the sample was made up in methanol.

The cycle time (including filling and equilibrating the CCC column, as well as sample injection, elution, and extrusion) was 90 min, as the extrusion of the column contents was performed with a fresh portion of the stationary phase at a high flow rate, so the column would be ready for the next injection.

The CCC fractions were collected every minute (Figure 1 and Appendix A) and pooled according to their chemical composition and purity, based on HPLC–UV, resulting in nine pooled fractions, three of which were later removed for further purification by preparative HPLC. The reproducibility of the CCC separation allowed for combining fractions, without additional analysis, of every single run, despite some variance in EEP sample composition. This variance could be avoided in the future by preparing a larger volume of a sample; thus, all injections would be made from the same sample solution.

### 2.2. UHPLC–MS Analysis, HPLC Purification

These pooled fractions were first analyzed by UHPLC–MS (Figure 2) to develop a specific gradient (based on their polarity) for further purification by preparative HPLC. The use of preparative HPLC instead of an open column guaranteed the reproducibility and the consistent throughput of the method.

The most nonpolar compounds eluted first in the fractions collected by CCC, beginning with the waxes, as expected (Appendix A). One of the compounds of interest which eluted with the CCC column volume at 32 min was an almost pure metabolite corresponding to Compound **1** (purity 96%, by HPLC–UV peak area). Its purification by preparative HPLC required an isocratic elution with 100% MeOH. The lack of commercial sources of this prenylated benzophenone and its biological activity potential makes the isolation process very valuable. The reddish fractions in Appendix A were eluted after step gradient switch. They correspond to more polar substances, including the red pigment of the resin. The column content contained almost pure red propolis pigment, or Compound **2**. It was observed that the polar (reddish colored) fractions contained most of the phenolic compounds. Their further purification by preparative HPLC led to the isolation of Compound **3**.

### 2.3. Structural Characterization of Purified Molecules

The deprotonated molecule of Compound **1**, of *m*/*z* 601.3516, was detected using the negative ionization mode (Figure 3b). It was purified by preparative HPLC, using 100% MeOH in isocratic conditions to eliminate some impurities, because the purity of the fractions obtained by CCC was 96%. It is a yellow substance present in most red propolis samples. This substance was assigned to a molecular formula of C_38_H_50_O_6_ and was tentatively assigned to any of the three isobaric benzophenones: isoxanthochymol, xanthochymol, or oblongofolin. Based on previous reports and also based on the fragments obtained by MS (Appendix A), oblongofolin was excluded, as it presents a fragment ion of *m*/*z* 399 [M − H]^−^, and its elution time LC is higher than that for xanthochymol or isoxanthochymol [27], which is in disagreement with our data. The nomenclature of some benzophenones varies according to the literature, and isoxanthochymol is considered to be the same molecule as guttiferone E [28]. Other studies describe xanthochymol and guttiferone E (or isoxanthochymol) as being an inseparable mixture [11]. At this point, it is suggested that the isolated compound with molecular formula C_38_H_50_O_6_ belongs to either xanthochymol or isoxanthochymol. Due to the nature of this purified fraction, it was not possible to use NMR for absolute identification, indicating that it could truly be an inseparable mixture.

Compound **2**, one of the main pigments present in red propolis, was mainly isolated from the extrusion process by CCC, with 88% purity. It is a red compound, and its protonated molecule of *m*/*z* 523.2852 was detected in the positive ion mode, leading to a molecular formula of C_32_H_26_O_7_. According to literature, a similar substance was identified by NMR and ESI-(+)-MS/MS [29]. The comparison of the fragmentation patterns of the data acquired in this study to that reported in the literature allowed for the identification of this compound as being retusapurpurin A (Figure 4). The molecular formula of these three substances and their main fragments was calculated considering a score above 90% and a mass error below 5 ppm (Table 4). The isotopic pattern was evaluated for each of the ions and compared to that of the calculated molecular formula for confirmation.

Finally, Compound **3** was detected as its deprotonated molecule of *m*/*z* 285.1141, [M − H]^−^, which was assigned a molecular formula of C_17_H_18_O_4_ and identified as (3*R*)-7-O-methylvestitol (Figure 4); its main fragments were also determined (Table 4). NMR was used to elucidate the structure, and H-NMR, HSQC, HMBC, COSY, and NOESY spectroscopy were acquired to identify this compound (Appendix A). All these results are in agreement with those previously reported in the literature for this substance [27].

## 3. Discussion

This study is a working example of CCC as a high-capacity separation technology for the fractionation of Brazilian red propolis at a preparative scale to produce enriched fractions for further purification by preparative HPLC. The combined use of a step gradient, flow rate modification after injection, and a sample made up in methanol, allowed for a better separation of the crude and better control of the stationary phase retention inside the column. The complexity of the extract of Brazilian red propolis required additional optimization of an earlier developed sample injection strategy.

Once the waxes were removed by CCC, the pooled fractions could be further purified by preparative HPLC, with a much higher loading of the sample without blocking the column. The structures of the isolated compounds were confirmed by NMR and high-resolution MS (Q-Tof) and were compared to data from the literature.

Only two other studies of CCC applied to propolis were found in the literature, but neither applied to red Brazilian propolis. Shahi et al., 2022 [30] combined CCC and deep eutectic solvent-based fibers for the extraction of antibiotic contaminants for the quality control of propolis from Iran. Green Brazilian propolis, which has a different plant source and composition, was submitted to CCC, followed by preparative chromatography, to isolate a known marker compound (Artepillin C) [31]. However, this is the first report of this combined strategy for the isolation of (not one, but three) components of red Brazilian propolis.

Based on NMR analysis, two out of nine pooled fractions collected contained a 96% pure prenylated benzophenone mixture of isoxanthochymol/xanthochymol (**1**) and 88% pure retusapurpurin A (**2**). These two pooled fractions, together with the most polar red fraction, were further purified by preparative HPLC, leading to the isolation of the isoflavan (3*R*)-7-O-methylvestitol (**3**).

These compounds were also detected in other studies of red Brazilian propolis from the State of Alagoas, Brazil, and incorporated in bio-polymeric hydrogel membranes to treat wounds [32]. The amount of propolis released from the three tested membranes was quantified via total phenol content, not by quantification of the compounds that were detected. One of the key problems in the evaluation of natural product activity is obtaining isolated compounds to carry out tests and to quantify these compounds in chromatographic analyses. Therefore, the use of CCC and preparative HPLC, as exemplified herein, would permit the isolation of these components directly from the sample matrix. Although the authors [32] acknowledge that some types of propolis cause allergic reactions, depending on their composition; they considered that this red propolis extract did not cause an allergic response.

Another study of red Brazilian propolis identified two different plant sources for this type of propolis [33], with *Dalbergia ecastaphyllum* as the botanical source of flavonoids such as liquiritigenin, medicarpin, and 7-O-neovestitol. A second source of plant resins, *Symphonia globulifera*, was identified as the source of polyprenylated benzophenones, mainly guttiferone E and oblongifolin B. These authors also observed that guttiferone E formed an inseparable mixture with its double-bond position isomer xanthochymol.

Many recent studies present important applications for this natural product, such as anti-*Helicobacter pylori* activity [34], but fail to consider the potential effect of variation in its composition. As red Brazilian propolis is based on (at least) two different plant sources, it is probable that its composition fluctuates, depending on the availability of resins, from either source, at the time of collection. Since these polyprenylated benzophenones are not commercially available standards, the method described herein for isolating nearly pure molecules (96%) is enticing and should be considered for the quality control of red Brazilian propolis-based products.

## 4. Materials and Methods

### 4.1. Materials

The red propolis sample was obtained from a beekeeper in the state of Alagoas (Brazil) in 2016 and extracted according to the method used by Sawaya et al. [10]. The sample was initially milled and then extracted by maceration for seven days in a flask, shaken at a rate of 100 rpm and at a temperature of 30 °C. A portion of 10 mL of absolute ethanol was used per 3 g of crude propolis. Subsequently, the insoluble part was filtered, and the filtrates were stored in the freezer at −16 °C overnight. To reduce the wax content, the extracts were filtered again at the same temperature. Finally, the solvent was removed using a rotary evaporator at 40 °C and 140 rpm to obtain the dried ethanol extracts of propolis.

The solvents used for CCC separations were of analytical grade, and for HPLC analysis, they were of HPLC grade, obtained from Fisher Chemicals (Loughborough, UK). HPLC-grade water was purified using a Purite Select Fusion Purewater system (Thame, UK).

### 4.2. Counter-Current Chromatography

The analytical scale CCC separations were performed on a DE-Mini system (Dynamic Extractions Ltd., Tafarnaubach, Wales) with a 17.4 mL PTFE column (0.8 mm I.D.). The rotation speed was 2100 rpm, and the flow was 1 mL·min^−^^1^. The system consisted of a Gilson 307 pump (ArtisanTG, Champaign, IL, USA) and a Knauer K2501PDA detector (KNAUER, Berlin, Germany). A PTFE sample loop of 0.43 mL was used, and the temperature of the CCC column system was 25 °C.

The CCC semi-purified fractions were obtained on a DE-Midi system (Dynamic Extractions Ltd., Wales), with two 4 mm I.D. polyfluoroalkoxy columns connected in series, yielding a total volume of 912 mL. The column was rotated at 1250 rpm and maintained at 25 °C. The DE-Midi was connected to an Agilent1200 preparative system (Agilent, Santa Clara, CA, USA). The CCC separation was conducted in normal phase elution mode (organic mobile phase) using a hexane:methanol:water (5:4:1 *v*/*v*/*v*) solvent system. The sample for injection was prepared in methanol. The column was first filled with a stationary phase (lower aqueous phase), then equilibrated at 40 mL·min^−1^. Just after sample injection, the flow was quickly reduced to 1.0 mL·min^−1^ for 10 min and then raised to 40 mL·min^−1^ until the end of the separation. At 35 min, a step gradient was applied by changing the mobile phase to the upper organic phase of hexane:ethyl acetate:methanol:water (1:1:1:1 *v*/*v*/*v*/*v*) for an additional 35 min. The separation was monitored at the following wavelengths: 210, 254, 280, and 390 nm. The fractions were collected using a Gilson-202 fraction collector every minute (40 mL fractions) after the initial 10 min.

### 4.3. Preparative HPLC

CCC fraction purification was performed on a Prominence preparative HPLC (Shimadzu, Kyoto, Japan) using LC-20 AT and a LC-6AD pumps, an SPD-20A UV detector, an SIL-20A auto-sampler for volumes up to 150 µL, or a manual injection with a loop of 2.0 mL for higher sample volumes, and a Gilson FC 203 B fraction collector. A preparative Varian Polaris 5 C18-ether column (250 × 21.2 mm I.D., 5 µm particle size) was used. Data acquisition was performed using LC Solutions software 1.25 from Shimadzu.

The analytical method developed for UHPLC–MS [6] was transferred to preparative chromatography and adapted for individual fractions. The maximum amount of the sample injected in each preparative chromatography run was determined experimentally (concentration varied between 40–150 mg·mL^−^^1^, depending on the composition of each fraction). The gradient used to isolate each compound varied according to their polarity, and the mobile phases used were methanol and water, with 0.1% formic acid (FA). The conditions for C_38_H_50_O_6_ (Compound **1**) were: the gradient starting at 90% methanol and increasing to 100% in 2 min, held until 17 min. The conditions for retusapurpurin A (Compound **2**) were: the gradient starting at 30% methanol, increasing to 40% in 2 min, then to 65% at 10 min, and to 75% at 16 min, and finally, to 90% at 19 min, held until 21 min before returning to the initial conditions, followed by column re-equilibration. The conditions for (3*R*)-7-O-methylvestitol (Compound **3**) were: the gradient starting at 40% methanol and held for 2 min, increasing to up 65% methanol at 15 min, increasing up to 80% at 17 min, and finally up to 90% at 19 min, held until 21 min before returning to the initial conditions, followed by column re-equilibration. The composition of the collected fractions was subsequently determined by UHPLC–MS.

### 4.4. UHPLC–MS

Compound identification was performed by UHPLC–MS, using a UPLC Acquity system coupled with a TQD Acquity mass spectrometer (Micromass-Waters Manchester, Manchester, UK), using an ESI source and a C18 BEH Waters Acquity column (2.1 mm × 50 mm × 1.7 µm particle size). The gradient started with 40% methanol, increasing to 100% in 9 min, held until 11 min, and then returning to the initial conditions, followed by column re-equilibration. Spectra were acquired using negative and positive ion modes under the following conditions: capillary of 3.00 kV, cone of 30.00 V, source temperature of 150 °C, desolvation temperature of 350 °C, and mass range between *m*/*z* 100–800.

### 4.5. High Resolution Mass Spectrometry ESI-QTOF–MS (QTOF)

The identification of the purified compounds was performed using an Agilent 6550 iFunnel QTOF mass spectrometer equipped with an ESI ionization source. The analyses were performed in either the positive or negative ion modes under the following conditions: capillary voltage of 3000 V, fragmentor voltage of 100 V, OCT 1 RF Vpp of 750 V; collision energy of 30 V, drying gas temperature of 290 °C; drying gas flow of 11 L·min^−^^1^; sheath gas temperature of 350 °C. The volume injected was 4 µL, employing a flow rate of 0.100 mL·min^−^^1^ using methanol and 0.1% FA.

### 4.6. Nuclear Magnetic Resonance (NMR)

To record NMR spectra of (3*R*)-7-O-methylvestitol, approximately 5 mg was dissolved in 0.6 mL of CDCl3. The NMR spectra were recorded on an Agilent DD2 spectrometer from Brazilian Biosciences National Laboratory (Brazilian Center for Research in Energy and Materials—CNPEM), operating at a 1H Larmor frequency of 499.726 MHz. Chemical shifts for protons were reported in parts per million (ppm) downfield from tetramethylsilane (TMS, 0 δ). Coupling constants (J) were given in hertz (Hz). Splitting patterns were designated as s, singlet; d, doublet; t, triplet; q, quadruplet; m, multiplet; dd, double doublet; ddd, double double doublet; dt, doublet of triplets. Carbon nuclear magnetic resonance spectra were recorded at 125.655 MHz, and chemical shifts for carbon were reported in parts per million (ppm) downfield from tetramethylsilane (TMS, 0 δ). The 2D homo- and heteronuclear spectra, such as COSY (nJH-H, scalar), NOESY (nJH-H, dipolar), and HSQC (1JH-C, scalar) e HMBC (nJH-C, scalar), were also evaluated.

## Figures and Tables

**Figure 1 molecules-29-02757-f001:**
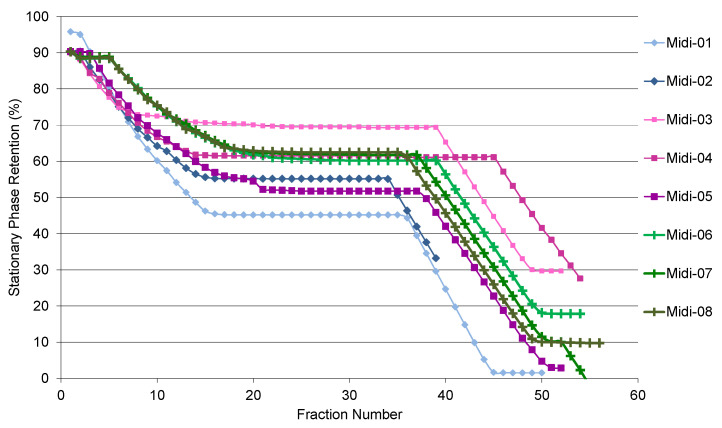
Effect of different loadings on stationary phase retention, according to the operating parameters described in Table 3.

**Figure 2 molecules-29-02757-f002:**
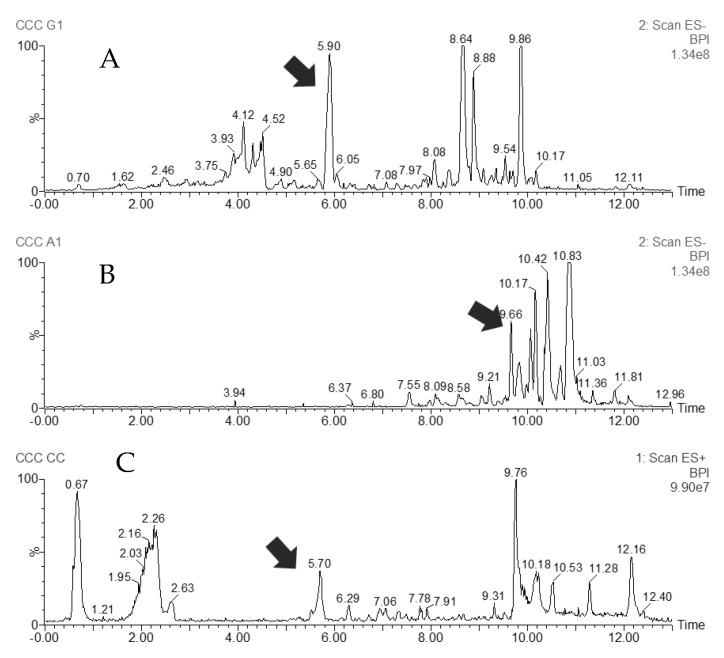
UHPLC–MS chromatograms of three pooled CCC fractions: (**A**) LC-ESI(-)-MS, a reddish fraction containing the most polar components, from which 3(*R*)-7-O-methylvestitol (Compound **3**) was isolated; (**B**) LC-ESI(-)-MS, an almost pure fraction of a prenylated benzophenone (Compound **1**); (**C**) LC-ESI(+)-MS, a red pigment, retusapurpurin A (Compound **2**), from extrusion fraction. Arrows indicate the target compound.

**Figure 3 molecules-29-02757-f003:**
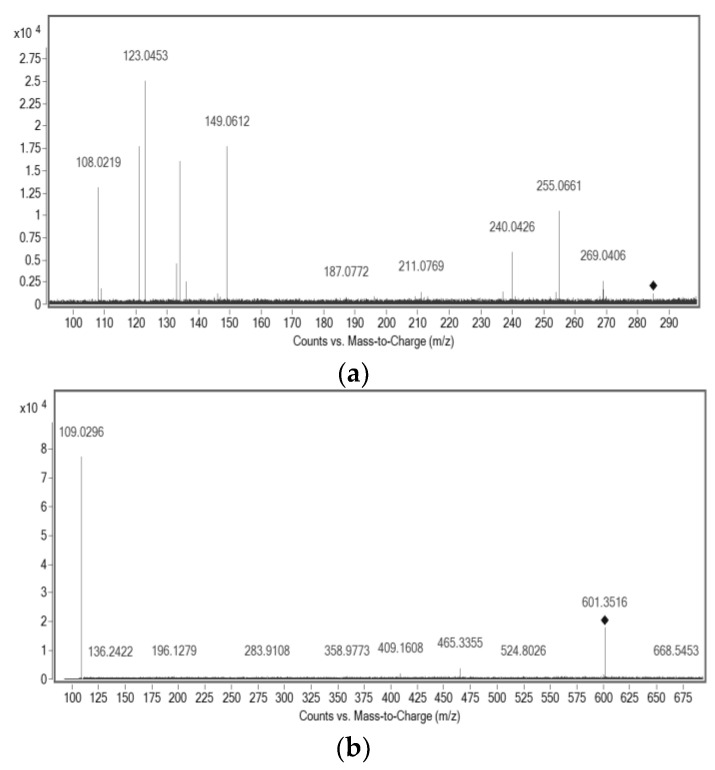
MS/MS spectra of the isolated compounds: (**a**) ESI-(-)-MS/MS of *m*/*z* 285.1141; (**b**) ESI-(-)-MS/MS of *m*/*z* 601.3516; (**c**) ESI-(+)-MS/MS of *m*/*z* 523.2852.

**Figure 4 molecules-29-02757-f004:**
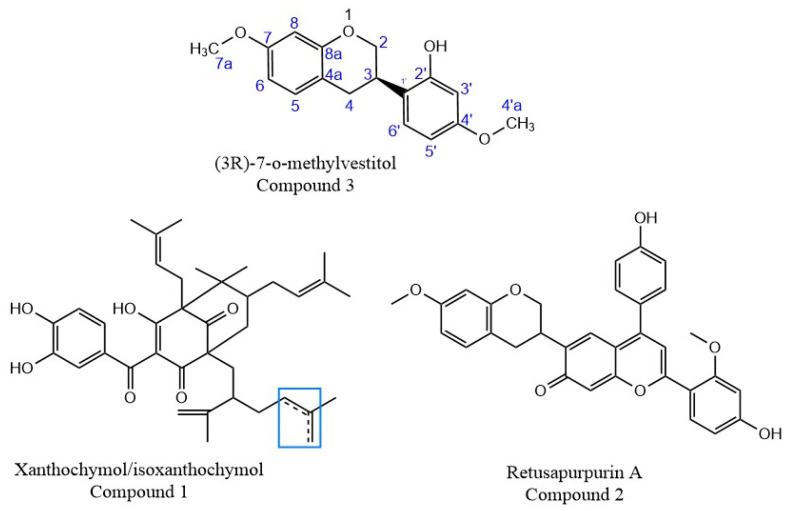
Chemical structures of the isolated compounds. The numbers in compound **3** (3*R*)-7-O-methyl correlate with the carbon numbers for the NMR analysis described in Appendix A (Appendix A). Box indicates region of isomerism.

**Table 1 molecules-29-02757-t001:** Solvent systems tested for red propolis extract fractionation and KD of the chosen marker compound for normal phase elution mode (solvent ratios are given as *v*/*v*/*v*/*v*).

Solvent	17	20A	23	25	26	27	2	3	4
Hexane	1	2.3	4	6	9	19	5	8	1
Ethyl acetate	1	1	1	1	1	1	0	0	0
Methanol	1	2.3	4	6	9	19	4	2	0
Water	1	1	1	1	1	1	1	0	0
Acetonitrile	0	0	0	0	0	0	0	5	1
K_D_ NP ^a^	0.1	0.1	0.6	2.0	3.3	5.6	1.2	17.5	16.7

^(a)^ Calculated as the ratio of the HPLC peak area in the lower phase to that in the upper phase.

**Table 2 molecules-29-02757-t002:** Partition coefficients (KD) ^a^ for the solvent systems chosen for the separation.

Retention Time (min)	Solvent System HMWat ^b^ (5:4:1)	Solvent System HEMWat ^c^ (1:1:1:1)
	LP/UP	LP/UP
2.466	5.9	21.9
2.739	30.2	37.1
4.014	353.0	286.2
4.311	362.1	2.8
4.901	742.9	3.1
5.200	509.3	2.2
5.615	760.0	1.2
5.841	618.3	1.2
6.009	546.4	1.1
6.160	511.3	0.8
6.279	279.9	0.9
6.520	108.9	1.1
6.626	684.0	0.0
6.786	290.6	1.2
6.972	180.5	2.2
7.079	140.8	0.0
7.335	4.1	0.7
7.479	451.5	1.0
7.719	6.3	0.0
8.132	29.2	0.6
8.453	165.6	0.3
8.690	29.1	0.0
9.278	0.7	0.1
9.919	128.6	0.1
10.759	4.4	0.0
10.974	3.0	0.4
11.836	57.2	0.0
13.075	2.7	0.0
14.080	0.1	0.0
14.418	1.2	0.1
14.985	0.6	0.0

^(a)^ Calculated as the ratio of the HPLC peak area in the lower phase (LP) to that in the upper phase (UP). ^(b)^ HMWat stands for hexane–methanol–water solvent system. ^(c)^ HEMWat stands for hexane-ethyl acetate–methanol–water solvent system.

**Table 3 molecules-29-02757-t003:** CCC operating parameters modified during loading studies using the step gradient of the mobile phase composition.

CCC Midi Run No.	Sample Solvent	Sample Volume (mL)	Sample Concentration (mg·mL^−1^)	Sample Loading (g)	Mobile Phase Flow Rate (mL·min^−1^)
1 ^a^	LP	20	100	2.00	50
2	MeOH ^b^	20	100	2.00	50
3	MeOH	10	100	1.00	40
4	MeOH	10	200	2.00	40
5	MeOH	10	400	4.00	40
6	MeOH	10	254	2.54	40
7	MeOH	10	254	2.54	40
8	MeOH	10	254	2.54	40

^(a)^ Run Midi-01 was performed at a constant flow of 50 mL·min^−1^. ^(b)^ MeOH stands for methanol.

**Table 4 molecules-29-02757-t004:** Molecular formula and mass error calculated for each precursor ion and their fragments.

Compound	Molecular Formula	*m*/*z* Experimental	*m*/*z* Theoretical ^a^		Mass Error (ppm)
**1**	C_38_H_50_O_6_	601.3522	601.3535	Precursor ion	1.78
	[C_6_H_5_O_2_]^−^	109.0296	109.0290	Product ion	−0.82
**2**	C_32_H_26_O_7_	523.2852	523.1727	Precursor ion	4.07
	[C_24_H_17_O_5_]^+^	385.0912	385.1037	Product ion	5.57
**3**	C_17_H_18_O_4_	285.1139	285.1132	Precursor ion	−2.85
	[C_7_H_7_O_2_]^−^	123.0453	123.0446	Product ion	−1.19
	[C_9_H_9_O_2_]^−^	149.0611	149.0603	Product ion	−2.23

^(a)^ Calculated using the isotopic distribution calculation from Masshunter.

## Data Availability

Data are available from the corresponding author on request.

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
