# Peer review of "Preparative Fractionation of Brazilian Red Propolis Extract Using Step-Gradient Counter-Current Chromatography"

_molecules, 2024, doi:10.3390/molecules29122757_

Round 1

Reviewer 1 Report

Comments and Suggestions for Authors

ARTICLE REVIEW

Title: Preparative fractionation of Brazilian red propolis extract by step-gradient counter-current chromatography

Authors: Begoña Gimenez-Cassina Lopez, Maria Cristina Marcucci, Silvana Aparecida Rocco, Maurício Luís Sforça, Marcos Nogueira Eberlin, Peter Hewitson, Svetlana Ignatova, Alexandra Christine Helena Frankland Sawaya

The work is well written.

In the introduction, the authors justified the selection of the research topic in detail.

In the chapter Results results obtained are presented in the form of 3 tables, 2 figures and supplementary materials (5 figures and 2 tables).

The obtained data were compared with the data of other researchers.

The Material and methods chapter describes the preparation of research material and describes in detail the research methods used in the work.

Unfortunately, the work does not include any conclusions resulting from the research conducted.

My comments:

Keywords: Expressions that are in the title of the article sholul not be repeared in keywords.

Line 134 - correct the record 17.4mL (no spece)

A uniform form of specifying units of measurement should be used throughout the work (mL/min or mL.min-1) – line 175, 192, 199, 209, 352, 362, 363, 379, 406, 408, 425 and table 3.

Line 358 – correct the record (25ºC.)

Line 425 – correct the record 40mg (no space)

In table 2 there is a dot at the end of the title. In the rest tables, none a dots.

Chapters: 2.2, 4.2, 4.3, 4.4, 4.5 and 4.6 should be justified.

Supplementary materials:

Figure S2. CCC chromatograms at different wavelengths. Conditions: Mini 17.4mL column, 0-25min HMWat (5:4:1), NP mode, 1 mL/min, 2100 rpm, 40mg of crude made in lower phase. – correct the record

Table S1 appears twice. No Table S2.

Author Response

REVIEWER 1

The work is well written.

In the introduction, the authors justified the selection of the research topic in detail. 

In the chapter Results results obtained are presented in the form of 3 tables, 2 figures and supplementary materials (5 figures and 2 tables). 

The obtained data were compared with the data of other researchers. 

The Material and methods chapter describes the preparation of research material and describes in detail the research methods used in the work. 

Unfortunately, the work does not include any conclusions resulting from the research conducted.

Response – we have included some of the supplementary material in the text and enriched the discussion to clarify how the method employed reached a practical purification of three bioactive molecules.

My comments:

Keywords: Expressions that are in the title of the article sholul not be repeared in keywords.

Response- The keywords have been modified: Injection strategy, Preparative fractionation, Retusapurpurin A, 3(R)-7-O-methylvestitol, Prenylated benzophenone isomers.

Line 134 - correct the record 17.4mL (no spece)

 Response – corrected 17.4 mL column

A uniform form of specifying units of measurement should be used throughout the work (mL/min or mL.min-1) – line 175, 192, 199, 209, 352, 362, 363, 379, 406, 408, 425 and table 3.

Response – corrected – all changed to mL.min-1

Line 358 – correct the record (25ºC.)

Response – corrected

Line 425 – correct the record 40mg (no space)

Response – corrected 40 mg

In table 2 there is a dot at the end of the title. In the rest tables, none a dots.

Response – corrected table 2 – no final period

Chapters: 2.2, 4.2, 4.3, 4.4, 4.5 and 4.6 should be justified.

Response – corrected – full text checked and justified.

Supplementary materials:

Figure S2. CCC chromatograms at different wavelengths. Conditions: Mini 17.4mL column, 0-25min HMWat (5:4:1), NP mode, 1 mL/min, 2100 rpm, 40mg of crude made in lower phase. – correct the record

Response –Figure S2 corrected.

CCC chromatograms at different wavelengths. Conditions: Mini 17.4 mL column, 0 – 25 min HMWat (5:4:1), NP mode, 1 mL.min-1, 2100 rpm, 40 mg of crude made in lower phase.

Table S1 appears twice. No Table S2.

Response- Table S1 has now been included in the text as Table 4. The supplementary material contains only one table (S1) together with the detailed discussion of the NMR attributions.

Reviewer 2 Report

Comments and Suggestions for Authors

Preparative fractionation of Brazilian red propolis extract by step-gradient counter-current chromatography

Begoña Gimenez-Cassina Lopez et al

Abstract

Propolis is a resinous bee product with a very complex composition which depends on the plant sources that bees visit. Due to the promising antimicrobial activities of red Brazilian propolis, it is paramount to identify the compounds responsible for them, which in most of the cases are not commercially available. The aim of this study was to develop a quick and clean preparative scale methodology for preparing fractions of red propolis directly from a complex crude ethanol extract by combining the extractive capacity of counter-current chromatography (CCC) with preparative HPLC. CCC method development included step gradient elution for the removal of waxes (which can bind to and block HPLC columns), sample injection in a single solvent to improve stationary phase stability and change of mobile phase flow pattern resulting in loading of 2.5g of the Brazilian red propolis crude extract on 912.5 mL Midi CCC column. Three compounds were subsequently isolated from the concentrated fractions by preparative HPLC and identified by NMR and high-resolution MS: red pigment, retusapurpurin A; the isoflavan, 3(R)-7-O-methylvestitol, and the prenylated benzophenone isomers, xanthochymol / isoxanthochymol . These compounds are markers of red propolis that contribute to its therapeutic properties and the amount isolated allows for further biological activities testing and as chromatographic standards.

The aim of this study is to develop a quick and efficient preparative-scale methodology for fractionating red Brazilian propolis directly from a complex crude ethanol extract by combining the extractive capacity of counter-current chromatography (CCC) with preparative HPLC.

I find this study interesting. However, it requires several improvements before publication:

1.      The English in the manuscript needs to be improved by a native speaker.

2.      The solvent ratios should be expressed as v/v/v/v, such as for HEMWat 17 (1:1:1:1). Please ensure consistency throughout the manuscript.

3.      The section on structure identification should be moved to the Results section and provide clearer elucidation of the structures.

4.      Figure 1 is of poor quality. Please redraw it using ChemDraw software.

5.      In the compound name "(3R)-7-O-methylvestitol," "R" should be italicized according to IUPAC rules. Check for consistency throughout the manuscript.

6.      Additional figure illustrations should be added to the Methods section to enhance clarity for the reader. I recommend including several in the supporting information adjacent to the Results section.

Comments on the Quality of English Language

Moderate editing of English language required

Author Response

REVIEWER 2

I find this study interesting. However, it requires several improvements before publication:

  1. The English in the manuscript needs to be improved by a native speaker.

Response- We would like to re-assure the Reviewer that at least two co-authors of this manuscript are native British English speakers. The language and the grammar have been carefully checked again.

  1. The solvent ratios should be expressed as v/v/v/v, such as for HEMWat 17 (1:1:1:1). Please ensure consistency throughout the manuscript.

Response- The expressions for the solvent system compositions are now consistent.

  1. The section on structure identification should be moved to the Results section and provide clearer elucidation of the structures.

And 6.      Additional figure illustrations should be added to the Methods section to enhance clarity for the reader. I recommend including several in the supporting information adjacent to the Results section

Response- Two figures (the UHPLC-MS chromatograms and the MS/MS fragmentation) were included in the results section and are now Figures 2 and 3, respectively. Furthermore, a table with the molecular formula and mass error of the three purified molecules has been brought into the

Reviewer 3 Report

Comments and Suggestions for Authors

This manuscript determines the fractionation of propolis extract using step-gradient counter-current chromatography. It is an interesting study and it is generally well written. Some minor comments to improve the readability are as below:

1) Introduction

Please italicize the scientific names.

2) Results & discussion

Discussion is slightly brief that requires more elaboration to explain the results obtained. 

Lack of clear justification on the solvent systems tested as tabulated in Table 1, based on what basis? Is there any targeted compounds from the beginning? Elaborate.

What the grey-coloured values (Table 2) indicate?

3) Most references are not so recent, please consider to look for more recent supporting literature.

Author Response

Reviewer 3

This manuscript determines the fractionation of propolis extract using step-gradient counter-current chromatography. It is an interesting study and it is generally well written. Some minor comments to improve the readability are as below:

1) Introduction

Please italicize the scientific names.

Response – this happened accidentally when we put the text into the Molecules format. We have gone through the text and italicized all scientific names.

2) Results & discussion

Discussion is slightly brief that requires more elaboration to explain the results obtained.

Response- Two figures (the UHPLC-MS chromatograms and the MS/MS fragmentation) were included in the results section and are now Figures 2 and 3, respectively. Furthermore, a table with the molecular formula and mass error of the three purified molecules has been brought into the main text (now Table 4). We agree that these modifications help to illustrate the identification of these molecules. However, we have left table S1 in the supplementary material, as well as the detailed description of the NMR assignments, which is probably of interest to a specific group only.

Furthermore, we have enriched the discussion to clarify how the method employed reached a practical purification of three bioactive molecules.

Lack of clear justification on the solvent systems tested as tabulated in Table 1, based on what basis? Is there any targeted compounds from the beginning? Elaborate.

Response- We have added additional reference [20] in the text above Table 1 for further clarification. Also, we have stated that “Several solvent systems (Table 1) from literature [22-24], as well as a standard approach used by the authors [20], were tested.”

What the grey-coloured values (Table 2) indicate?

Response- The highlighting was accidently left in the table and it has been removed. Thank you for pointing it out.

3) Most references are not so recent, please consider to look for more recent supporting literature. 

Response- One would think there would be many recent papers, but there are surprisingly few papers dealing with propolis and CCC! The only two new ones that were found have now been included the in the discussion (Shahi et al. 2022 and Arruda et al., 2020). Neither of them report findings on this type of propolis, red Brazilian propolis, so this study is the first!

Round 2

Reviewer 2 Report

Comments and Suggestions for Authors

The authors made substantial revisions based on my comments. I believe it is now suitable for publication

Comments on the Quality of English Language

Minor editing of English language required